# Micronutrient Status of Recreational Runners with Vegetarian or Non-Vegetarian Dietary Patterns

**DOI:** 10.3390/nu11051146

**Published:** 2019-05-22

**Authors:** Josefine Nebl, Jan Philipp Schuchardt, Alexander Ströhle, Paulina Wasserfurth, Sven Haufe, Julian Eigendorf, Uwe Tegtbur, Andreas Hahn

**Affiliations:** 1Faculty of Natural Sciences, Institute of Food Science and Human Nutrition, Leibniz University Hannover, 30159 Hannover, Germany; nebl@nutrition.uni-hannover.de (J.N.); schuchardt@nutrition.uni-hannover.de (J.P.S.); stroehle@nutrition.uni-hannover.de (A.S.); wasserfurth@nutrition.uni-hannover.de (P.W.); 2Institute of Sports Medicine, Hannover Medical School, 30625 Hannover, Germany; Haufe.Sven@mh-hannover.de (S.H.); Eigendorf.Julian@mh-hannover.de (J.E.); Tegtbur.Uwe@mh-hannover.de (U.T.)

**Keywords:** vegetarianism, veganism, recreational athletes, nutrient supply, nutrient status

## Abstract

Vegetarian diets have gained popularity in sports. However, few data exist on the status of micronutrients and related biomarkers for vegetarian and vegan athletes. The aim of this cross-sectional study was to compare the micronutrient status of omnivorous (OMN, *n* = 27), lacto-ovo-vegetarian (LOV, *n* = 26), and vegan (VEG, *n* = 28) recreational runners. Biomarkers of vitamin B_12_, folate, vitamin D, and iron were assessed. Additionally, serum levels of calcium, magnesium, and zinc were examined. Lifestyle factors and supplement intake were recorded via questionnaires. About 80% of each group showed vitamin B_12_ adequacy with higher levels in supplement users. Mean red blood cell folate exceeded the reference range (>340 nmol/L) in all three groups (OMN: 2213 ± 444, LOV: 2236 ± 596, and VEG: 2354 ± 639 nmol/L; not significant, n.s.). Furthermore, vitamin D levels were comparable (OMN: 90.6 ± 32.1, LOV: 76.8 ± 33.7, and VEG: 86.2 ± 39.5 nmol/L; n.s.), and we found low prevalence (<20%) of vitamin D inadequacy in all three groups. Less than 30% of each group had depleted iron stores, however, iron deficiency anemia was not found in any subject. Our findings suggest that a well-planned, health-conscious lacto-ovo-vegetarian and vegan diet, including supplements, can meet the athlete’s requirements of vitamin B_12_, vitamin D and iron.

## 1. Introduction

Micronutrients such as vitamins as well as major and trace minerals are involved in various metabolic processes important to physical performance [1]. Regular physical activities are associated with several biochemical training adaptations like an increased expression of antioxidant enzymes or increased blood formation, which, as a result, cause higher micronutrient requirements. Moreover, studies dealing with competitive athletes observed insufficient intake of energy-supplying macronutrients but also micronutrients, especially in athletes with unfavorable or restricted food choices [2,3]. Additionally, due to increased physical stress, through increased sweating and losses via urine, feces, and foot-strike hemolysis, athletes might have increased requirements of several micronutrients like iron and zinc. There is well-defined research that underlines the fact that an inadequate micronutrient status compromises physical performance and regeneration capacity, whereas the risk of upper respiratory tract infections (URTI) increases [4,5,6].

Currently, plant-based nutrition is a topical issue in sports medicine. There is debate about whether a plant-based diet can provide all the required nutrients in adequate amounts for an athlete. However, several nutrition societies have concluded that well-planned vegetarian dietary patterns, including a wide variety of plant foods, can be adequate for athletes [7,8,9,10,11,12], and numerous sportspeople from various disciplines have already shifted to a plant-based diet [13].

Previous studies have examined the nutritional status of vegans compared to vegetarians and omnivores, but these studies were largely with non-athletes [14]. In general, individuals following a plant-based diet showed adequate supply with most nutrients. However, it is generally assumed that limiting animal products in the diet increases the risk of certain micronutritional deficiencies. Actually, “the more foods eliminated from any diet, the greater the risk of deficiency” [15]. Undoubtedly, animal-derived foods like lean red meat, fish, and eggs are excellent sources of vitamin B_12_ and provide high amounts of bioavailable zinc, iron, and vitamin D [16,17,18]. Furthermore, dairy products are rich in calcium and other minerals [19]. Consequently, calcium, zinc, iron, vitamin B_12_, and vitamin D are described as critical nutrients in vegetarian and especially vegan dietary patterns [20].

However, up to now, only few data on the micronutrient biomarker status of vegetarian athletes exist [21,22]. Furthermore, most studies did not differentiate the various types of vegetarianism. Thus, the micronutrient status of athletes consuming a vegetarian and vegan dietary pattern is rather unknown.

To fill this knowledge gap, the approach of this study was to determine the nutritional status of selected parameters of vegan recreational runners in comparison to lacto-ovo-vegetarians and omnivores. In addition, we ascertained the influence of dietary supplement use on micronutrient biomarker status. It was hypothesized that micronutrient status differed between the groups.

## 2. Materials and Methods

### 2.1. Study Design and Participants

This cross-sectional study was conducted according to the guidelines laid down in the Declaration of Helsinki. The ethics committee at the medical chamber of Lower Saxony (Hannover, Germany) approved all procedures. All subjects gave their written informed consent. The study was registered in the German Clinical Trial Register (DRKS00012377).

The study was conducted at the Institute of Food Science and Human Nutrition, Leibniz University Hannover, Germany.

Eighty-one healthy omnivorous (OMN), vegetarian (LOV), and vegan (VEG) recreational runners (men and women) aged between 18 and 35 years were recruited from the general population in Hannover, Germany, via local running events, online running communities, as well as online vegetarian and vegan communities.

Eligibility of subjects was assessed using questionnaires. Subjects were preselected via screening questionnaires according to the following inclusion criteria: omnivorous, lacto-ovo-vegetarian, or vegan diet for at least half a year, body mass index (BMI) between 18.5 and 25.0 kg/m^2^, and regularly running (2 to 5 times per week) for at least 30–60 min. Regular running sessions were documented via self-reporting data. The following criteria led to exclusion: any cardiovascular, metabolic, or malignant disease; diseases regarding the gastrointestinal tract; pregnancy; food intolerances; and addiction to drugs or alcohol. The use of dietary supplements did not lead to exclusion except if they were performance-enhancing substances (e.g., alkaline salts, creatine).

The categorization of omnivorous, lacto-ovo-vegetarian, and vegan was based on questionnaires, which initially included a question about the current diet. Secondly, consumed food groups were queried to make sure that the participants classified themselves correctly. Subjects were classified as “omnivorous” if they consumed grains, plant foods, legumes, milk and dairy products, and eggs as well as fish, meat, and meat products. “Lacto-ovo-vegetarians” were characterized by the consumption of grains, plant foods, legumes, milk and dairy products, and eggs. The consumption of grains, plant foods, and legumes characterized “vegans”. Participants that were included in the study population were matched according to age and gender. They were invited to the Institute of Food Science and Human Nutrition of the Leibniz University Hannover for a comprehensive examination. Participants completed a questionnaire regarding their supplement intake (frequency and dosage), health status, and running activity. Training frequency and duration were self-reported by the subjects.

### 2.2. Analytical Methods

After overnight fasting (≥10 h fasting period), blood samples were collected between 06:00 and 10:00 a.m. Blood samples were obtained by venipuncture of an arm vein using Multiflyneedles (Sarstedt, Nümbrecht, Germany) into serum, EDTA, or special monovettes for tHcy (Sarstedt). All samples were stored at ~5 °C and were transferred to the laboratory on the same day.

All micronutrient parameters described below were determined in an accredited and certified laboratory (Laborärztliche Arbeitsgemeinschaft für Diagnostik und Rationalisierung e.V., in Hannover, Germany).

Briefly, vitamin B_12_ and holotranscobalamin (Holo-TC) in serum were determined with the use of the electrochemiluminescence immunoassay method (ECLIA) on cobas® test systems (Roche Diagnostics GmbH, Mannheim, Germany) according to [23,24] (pp. 281–283). Liquid chromatography with mass spectrometry coupling (LC-MS/MS) was applied to assess methylmalonic acid (MMA) in serum [25]. Plasma homocysteine (tHcy) was determined by HPLC with a fluorescence detector in accordance with [26]. The four marker combined vitamin B-12 indicator (4cB12) was computed from concentrations of vitamin B_12_ in serum, Holo-TC, MMA, and homocysteine according to published formula [27]:
$$4\mathrm{cB}12=\log_{10}\left(\frac{{\mathrm{HoloTC}*B}_{12}}{\mathrm{MMA}*\mathrm{tHcy}}\right)-\left(\mathrm{age}\;\mathrm{factor}\right).$$


Vitamin D status (25-hydroxyvitamin D, 25(OH)D) was measured in serum by ECLIA (Roche Diagnostics GmbH, Mannheim, Germany) according to [28].

Folate was analyzed in red blood cells (RBCs) as a reliable biomarker for tissue folate status [29] via ECLIA on cobas® 8000 modular analyzer series (Roche Diagnostics GmbH, Mannheim, Germany) [30].

Calcium status [31] and magnesium [32] status were assessed in serum by a photometric method (Beckman Coulter®, Krefeld, Germany). Atomic absorption spectrometry (AAS, PerkinElmer, Inc., Waltham, Massachusetts) was used to measure serum zinc concentrations [33]. Iron in serum was assessed using a photometric method according to [34]. To determine iron stores, ferritin was measured via immunoturbidimetric assay (Beckman Coulter®, Krefeld, Germany) as well as transferrin in serum [35,36]. A photometric method was used to measure hemoglobin (Hb) [37,38,39]. Transferrin saturation [40], hematocrit (Hct) [41], and mean corpuscular volume (MCV) [42] were calculated using standard formulas.

### 2.3. References Values

According to a WHO Technical Consultation on vitamin B_12_ deficiencies, a cutoff value for serum vitamin B_12_ was set at <150 pmol/L [43] to indicate deficiency, and cutoffs for related parameters were set as the following: MMA > 271 mmol/L [44,45], Holo-TC < 35 pmol/L [46], and homocysteine > 10 µmol/L [47,48].

The following five categories for the dimensionless unit score 4cB12 were classified: probable vitamin B_12_ deficiency (<−2.5), possible vitamin B_12_ deficiency (−2.5 to −1.5), low vitamin B_12_ (−1.5 to −0.5), vitamin B_12_ adequacy (−0.5 to 1.5), and elevated vitamin B_12_ (>1.5) [27].

Vitamin D status was assessed according to the following 25(OH)D thresholds: <25.0 (deficiency), 25.0 to 49.9 (insufficiency), 50.0 to 74.9 (sufficiency), and ≥75.0 nmol/L (optimal) [49,50,51].

According to the WHO Consultation, RBC folate levels <340 nmol/L were regarded as deficient [43].

The following cutoff points were used to define adequate biomarker status: calcium 2.2–2.6 mmol/L [52] (pp. 231–234), magnesium 0.65–1.05 mmol/L [53] (pp. 339–340), and zinc 12–15 µmol/L [54]. According to the WHO, cutoffs for parameters of iron status were set as: serum iron <10 µmol/L, ferritin <15 µg/L (depleted iron stores), Hb <13 or <12 g/dL (men and women, respectively, anemia), MCV <80 fl (indication for iron deficiency anemia), transferrin ≥47.7 g/L (increased iron requirement), and transferrin saturation <16% (insufficient iron supply) [41].

### 2.4. Data Analysis and Statistical Methods

Data are shown as mean ± standard deviation (SD) or frequency and percent. SPSS software (IBM SPSS Statistics 24.0; Chicago, IL, USA) was used for statistical analyses. The Kolmogorov–Smirnov test was used to control distribution. If data were normally distributed, a one-way analysis of variance (ANOVA) was used to evaluate differences in nutritional status between the three diet groups. In contrast, the Kruskal Wallis test was performed to analyze data with non-normal distribution. Afterwards, a post hoc test with Bonferroni correction was conducted to analyze differences between the individual groups. In order to examine differences between supplement users (SU) and non-SU or men and women within the groups, the t-test (for parametric data) and the Mann–Whitney U test (for nonparametric data) were used. Moreover, to compare the differences between the frequencies of the three groups, a chi-square test was used. In addition, Pearson correlation was computed to calculate correlations between parametric data. Finally, to assess associations between nonparametric data, Spearman´s rho correlation was used. Statistical significance was set at the 0.05 level.

## 3. Results

In total, 27 OMN, 26 LOV, and 28 VEG met the inclusion criteria and were included in the study. Between the three groups, there were no differences regarding gender distribution as well as mean in age and BMI (Table 1). All three groups consumed comparable frequencies of dietary supplements, except vitamin B_12_, which was the most commonly used supplement among VEG. LOV and VEG followed their diet for a shorter period compared to OMN (*p* = 0.001, χ^2^). All participants were nonsmokers and had similar training habits (Table 1).

### 3.1. Biomarkers of Vitamin B_12_ Status

Overall, all three groups showed an adequate biomarker status of vitamin B_12_-related parameters (Table 2), even when considering only men or women as subgroups (Appendix A). However, the vitamin B_12_ status of supplement users of VEG and OMN was higher compared to non-SU, and a higher proportion of the non-SU had B_12_ parameters outside the reference range.

Regarding 4cB12, on average, all three groups had an adequate status (Figure 1). Most subjects of each group (~80%) showed vitamin B_12_ adequacy, while 19% of LOV, 16% of OMN, and 7% of VEG had an elevated vitamin B_12_ status (>1.5). Again, the vitamin B_12_ biomarker status was higher in SU (statistically significant within the VEG group), and an overall tendency (significant for non-SU in LOV and VEG) towards lower levels in VEG compared to OMN and LOV was observed (Figure 1).

An inverse association was observed for homocysteine and Holo-TC in all three groups (r = −0.505, *p* = 0.007; r = −0.720, *p* < 0.001; and r = −0.400, *p* = 0.035 for OMN, LOV, and VEG, respectively). Also, inverse associations were found in the VEG group between homocysteine, vitamin B_12_ in serum (r = −0.577, *p* = 0.002), and MMA (r = 0.430, *p* = 0.028). The 4cB12 level was positively correlated with the average daily vitamin B_12_ supplement intake in the VEG group (r = 0.422, *p* = 0.025) but not in the other two groups. Further, there was a tendency of lower 4cB12 in non-SU of the VEG group, who had been vegan for a longer period compared to those who practiced a vegan diet for a shorter period (*p* = 0.039).

### 3.2. Biomarkers of Folate Status

RBC folate status exceeded the reference range (>340 nmol/L) in all three groups (Table 3). Additionally, in LOV, RBC folate levels were positively associated with serum vitamin B_12_ (r = 0.494, *p* = 0.010) and Holo-TC levels (r = 0.629, *p* = 0.001), while no correlations could be observed in the VEG and OMN groups. No significant differences were found in SU compared to non-SU in OMN (*p* = 0.865) and LOV (*p* = 0.198), but significance was found for VEG (*p* = 0.030).

### 3.3. Biomarkers of Vitamin D Status

Average values of 25(OH)D were in the reference range in all three groups without any differences between the dietary groups (Table 3) and also according to gender (Appendix A). However, with 76.8 nmol/L the 25(OH)D levels of LOV were in the lower reference range. An inadequate vitamin D biomarker status (<50 nmol/L) was found in <25% of all three groups. Again, vitamin D biomarker status was dependent on supplementation. In detail, all participants who took vitamin D supplements had values >50 nmol/L (Table 3). Furthermore, 25(OH)D concentrations of the total non-SU population were significantly higher in summer (83.1 ± 30.7 nmol/L) than in winter (66.0 ± 28.0 nmol/L; *p* = 0.021) and, regardless of the season, they were higher in SU (124 ± 30.1 nmol/L) compared to non-SU (76.9 ± 30.7 nmol/L; *p*< 0.001). The 25(OH)D level was positively associated with the average daily vitamin D supplement intake in OMN (r = 0.400, *p* = 0.039) and VEG (r = 0.576, *p* = 0.001) but not in LOV (n.s.).

### 3.4. Biomarkers of Iron Status and Hematological Parameters

Hematological and iron status parameters are shown in Table 4. On average, ferritin concentrations were in the reference range for all three dietary groups and both genders. Considering only men, OMN showed significantly higher ferritin concentrations (115 ± 44.8 µg/L) compared to LOV (64.3 ± 40.8 µg/L, *p* = 0.024) and VEG (64.6 ± 36.7 µg/L, *p* = 0.028) (Table 4). There were no differences among OMN, LOV, and VEG women (25.9 ± 23.0 vs. 23.9 ± 12.9 vs. 32.1 ± 22.8 µg/L, n.s., respectively). In all three groups, significantly higher levels of most parameters were found in men. Depleted iron stores (ferritin <15 µg/L) were observed only in women (26% of OMN, 23% of LOV, and 18% of VEG) without significant differences between the groups (*p* = 0.619). Biomarkers of iron status were not associated with iron supplement intake in any group.

Intake of oral contraceptives was observed in 63% of female OMN, 31% of LOV, and 11% of VEG, but no associations to parameters of iron metabolism were found (n.s.). In addition, no significant differences were observed between iron SU and non-SU (Appendix A).

### 3.5. Serum Levels of Calcium, Zinc, and Magnesium

Independent of supplementation, calcium (OMN: 2.45 ± 0.09, LOV: 2.45 ± 0.07, and VEG: 2.45 ± 0.10 mmol/L), zinc (OMN: 14.1 ± 1.82, LOV: 13.5 ± 2.50, and VEG: 12.3 ± 2.17 µmol/L), and magnesium (OMN: 0.83 ± 0.05, LOV: 0.83 ± 0.05, and VEG: 0.86 ± 0.06 mmol/L) serum levels were in the reference range in all groups, and no subject had calcium or magnesium levels below the reference range. Regarding zinc, 50.0% of VEG, 23.1% of LOV, and 11.1% of OMN had low levels (<12 µmol/L).

## 4. Discussion

This is the first cross-sectional study evaluating the biomarker status of several vitamins and minerals of recreational runners practicing vegetarian and vegan diets compared to an omnivorous diet. Since there were no comparable values of vegan recreational runners in the literature, our study results were compared with data of vegan/vegetarian nonathletes as well as with omnivorous athletes.

### 4.1. Vitamin B_12_

There are different biomarkers for assessing the vitamin B_12_ status including vitamin B_12_ and Holo-TC in serum or plasma as well as methylmalonic acid and homocysteine in serum. In addition, several cutoff values were defined to assess the vitamin B_12_ status [55]. Plasma vitamin B_12_ concentration by itself does not reliably unveil vitamin B_12_ deficiency and should, therefore, be determined in combination with functional vitamin B_12_ parameters such as MMA [56]. In contrast to vitamin B_12_ levels in plasma, circulating Holo-TC represents the most sensitive parameter for diagnosing early vitamin B_12_ deficiency [57]. Elevated MMA and tHcy levels in plasma are also sensitive metabolic markers for low vitamin B_12_ levels [20]. However, the specificity of MMA and tHcy as biomarkers for vitamin B_12_ status is limited [58]. As a novel score to determine the vitamin B_12_ status, the 4cB12 indicator combines serum vitamin B_12_, serum Holo-TC, plasma tHcy, and serum MMA [27]. The formula for calculating 4cB12 was established with a database of 5211 subjects from various nations, ages, and health status [27].

In our study, all vitamin B_12_ biomarkers showed an adequate to optimal supply when compared to reference values. The findings were independent of the respective dietary group but dependent on supplementation. Considering the non-SU, it was obvious that VEG had the lowest vitamin B_12_ supply markers although in an adequate supply area. Interestingly, there were only marginal differences between the non-SU of the OMN and LOV groups, suggesting that a lacto-ovo-vegetarian diet, containing milk and dairy products, provides certain amounts of vitamin B_12_. Considering the duration of the diet, no impact on the vitamin B_12_ supply in OMN and LOV was found. Nevertheless, our results should be considered with caution, as these findings are in contrast to most studies showing an inadequate B_12_ status in the majority of vegans and vegetarians, as summarized in [56]. For example, in systematic reviews, an inadequate vitamin B_12_ serum status was observed in up to 87% of vegetarians and vegans, and this was combined with elevated MMA levels (32%–83%) and decreased Holo-TC levels (72%–93%) [56,59]. However, some studies included only non-SU or did not differentiate regarding vitamin B_12_ supplementation [56,59]. But, the Adventist Health Study 2 clearly showed an association between vitamin B_12_ supplementation and serum B_12_ and Holo-TC levels [60].

Despite the adequate supply, increased tHcy levels (>10 μmol/L) were found in >50% of all three dietary groups. The observed tHcy levels (OMN: 12.2 ± 2.93, LOV: 13.2 ± 6.47, and VEG: 12.8 ± 4.26 µmol/L) partly agreed with data from a meta-analysis, which found average tHcy levels of 16.4 µmol/L in vegans, but, in contrast to the present results, this study found significantly higher tHcy in vegans compared to omnivores and vegetarians [48].

Homocysteine is an independent risk factor for ischemic heart disease and ischemic stroke. By increasing the homocysteine concentration by 5 µmol/L, the risk of coronary events increases by 18%, and the risk of stroke increases by 19% [58]. Therefore, recreational athletes adopting a vegan diet should be encouraged to take dietary vitamin B_12_ supplements.

### 4.2. Red Blood Cell (RBC) Folate

Red blood cell folate concentration is a sensitive biomarker for the folate status, and concentrations <340 nmol/L are indicative of folate deficiency in healthy adults [43]. In contrast, cutoff values of ≥906 nmol/L have been estimated for women of reproductive age for the prevention of neural tube effects. In the present study population, high RBC folate concentrations were observed, and the mean levels of folate concentrations in the erythrocytes were similar in all three dietary groups (OMN: 2213 ± 444, LOV: 2236 ± 596, and VEG: 2354 ± 639 nmol/L; p = 0.577). Results of non-SU of the three groups revealed a very homogenous picture. Since in Germany foods are not fortified with folate (with the exception of salt), the results suggest a high intake of folate-rich foods, like green leafy vegetables, because of the high health compliance in all three groups.

As the values appeared to be extraordinarily high, the applied method was tested and internally validated on the basis of several laboratories (LabA, LabB, and LabC). The results of the comparisons of the laboratories showed a clear correlation between the present values and the values of other laboratories (r_LabA-LabB_ = 0.656, *p* = 0.039; r_LabA-LabC_ = 0.902, *p* < 0.001). Currently, other studies have found comparably high RBC folate levels. A recent study by Gallego-Narbón found average RBC folate levels of 1704 nmol/L in Spanish lacto-ovo-vegetarians and vegans [61]. Also, a recent publication of the Adventist Health Study 2 showed RBC folate levels >2000 nmol/L [62].

### 4.3. 25-Hydroxyvitamin D (25(OH)D)

The serum 25(OH)D concentration reflects the amount of vitamin D attained from both dietary sources and endogenous synthesis and is a sensitive biomarker for the vitamin D status. However, there is no final consensus about the concentration of serum 25(OH)D that would achieve the greatest benefit for health. While 25(OH)D levels of >75 nmol/L are considered as optimal, levels ≥50 nmol/L are regarded as adequate, levels of 25–49.9 nmol/L as inadequate, and levels <25 nmol/L as deficient [51,63]. The difficulty in assessing the vitamin D status is more due to the methodology, as ECLIA was discussed to overestimate the 25(OH)D levels [64]. Consequently, in addition to ECLIA, 25(OH)D levels were analyzed via liquid chromatography tandem-mass spectrometry (LC-MS/MS) [65] for validation as well. The variability was <10%, which was considered tolerable.

In our study, all three dietary groups had comparable average 25(OH)D values of >75 nmol/L and showed a similar low prevalence (<20%) of vitamin D inadequacy (<50 nmol/L), while vitamin D deficiency (<25 nmol/L) was present in only two VEG subjects. Nevertheless, the vitamin D supply of the present population is to be regarded as exceptional, since there is a high prevalence of vitamin D deficiency in the general population [63]. Adequate vitamin D supply in the present subjects could be explained by the high proportion of SU (specific vitamin D supplements as well as multivitamin supplements). Interestingly, all subjects who consumed vitamin D (containing) supplements had 25(OH)D levels >50 nmol/L and, therefore, had an adequate supply, while inadequate to deficient supply concerned only non-SU. Overall, in the total study population the vitamin D status was dependent on vitamin D supplement intake.

Our findings are consistent with results of the Adventist Health Study 2, which showed that the vitamin D status depends only to a small extent on nutritional factors and much more on supplementation [66]. Additionally, a meta-analysis showed that only about 56% of 2313 athletes (22.5 ± 5 years old) had inadequate vitamin D levels (defined by the authors with <80 nmol/L), which was influenced by season, type of sport (indoor/outdoor), and latitude [67]. These results are largely consistent with the present findings, where 33% of OMN, 43% of VEG, and 58% of LOV were not optimal supplied (<75 nmol/L).

There are many factors which could influence vitamin D status. Presumably, the adequate supply status is due to the relatively long stay outdoors, because other studies also showed an adequate supply status of female runners [68]. All subjects were recruited from Hanover or the surrounding area, which has a latitude of 52°N. In contrast to the present results, a German nationwide (latitude of 52°–54°) study with almost 7000 subjects (age 18–79 years) showed 25(OH)D levels of 45.1 and 45.3 nmol/L for men and women, respectively [69]. Since the present study population was about 27 years old, it could be assumed that there was sufficient intrinsic synthesis [69]. Additionally, our examinations took place from summer to winter months (May to December 2017), so a high endogenous synthesis could be expected [70]. Additional influences of vitamin D status such as recent holidays, individual sun exposure, as well as sun protection habits (e.g., the use of sun cream or sun-protecting clothes) [71] were not examined.

### 4.4. Iron

There is wide consensus that serum ferritin is the most sensitive parameter to detect depleted iron stores and, therefore, isolated iron deficiency [40,72,73]. Depleted iron stores are present if ferritin concentrations are <15 µg/L [72]. However, in addition to depleted iron stores, iron deficiency anemia can only be diagnosed in the presence of decreased serum iron, transferrin saturation, Hb, and MCV levels [40].

Considering the ferritin levels of the present subjects, <30% of each group had depleted iron stores, but iron deficiency anemia was not found in any subject. Surprisingly, female VEG had the highest mean ferritin levels (32.1 ± 22.8 µg/L), while among men the OMN group showed the highest levels (115 ± 44.8 µg/L). Again, high interindividual variations were observed. In contrast to parameters of vitamin B_12_ and vitamin D metabolism, supplementation was not crucial for an adequate iron supply—also found for LOV and VEG. Since the iron bioavailability is higher in animal-based foods compared to plant-based foods (10%–20% vs. 1%–5%), a similar status could be only achieved through a high intake of iron-containing plant-based foods such as whole grain and legumes as well as availability-enhancing food ingredients (e.g., vitamin C). Therefore, our data indicate that a targeted choice of plant foods can also ensure an adequate iron supply.

Our results are largely consistent with the literature, since vegetarians and vegans, in general, show hemoglobin and serum iron levels in the reference range. Also, iron deficiency is not more common than in omnivores [14,74,75], whereas ferritin levels, in contrast to our results, are often low [76,77]. For example, Elorinne and colleagues observed significantly lower ferritin levels in vegans (median of 26 µg/L) compared to nonvegetarians (median of 72 µg/L) [75]. Moreover, even 8% of female vegans of the German Vegan Study had iron deficiency anemia [77]. Therefore, especially in female recreational athletes who consume (almost) only plant-based foods, the risk of insufficient iron supply is increased [78,79].

### 4.5. Calcium, Zinc, and Magnesium

Average values of all three groups showed adequate serum levels of all three minerals. Most subjects of VEG had low levels of zinc, which was consistent with previous findings [14]. However, in contrast to the previously mentioned parameters, serum levels of calcium, zinc, and magnesium are not directly linked to dietary intake as a result of the tight homeostatic regulation of blood levels.

### 4.6. Limitations

In addition to the small sample size, only a certain age group (18–35 y) was examined. Also, the study took place mainly in summer months, which possibly influenced study results. Further, the current training phase could have an effect on the results.

### 4.7. Future Research Directions

Future research is needed to examine the nutrient status of vegetarian/vegan athletes. Also, data of various disciplines and different levels (e.g., professional level) would be of great interest. In addition, intervention studies are needed to investigate the influence of a vegetarian/vegan diet on various biomarkers.

## 5. Conclusions

In summary, our data suggest that a well-planned, health-conscious lacto-ovo-vegetarian and vegan diet, including supplements, can meet the recreational athlete’s requirements of vitamin B_12_, vitamin D, and iron.

## Figures and Tables

**Figure 1 nutrients-11-01146-f001:**
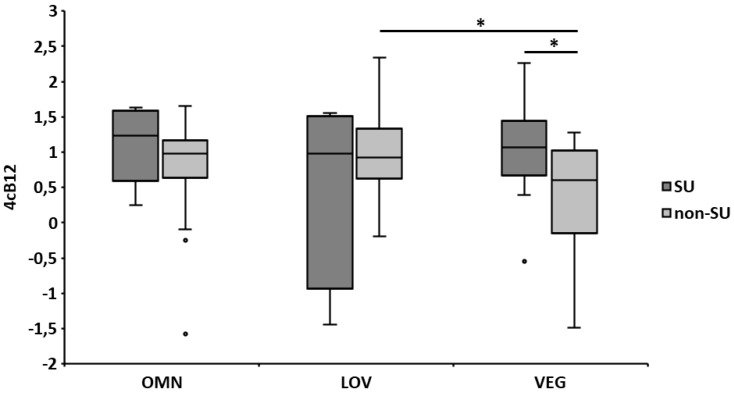
Vitamin B_12_ indicator (4cB12) of the dietary patterns according to supplementation. Categories of vitamin B_12_ status: <−2.5 = probable B_12_ deficiency, −2.5 to −1.5 = possible B_12_ deficient, −1.5 to −0.5 = low vitamin B_12_, −0.5 to 1.5 = B_12_ adequacy, and >1.5 = elevated B_12_ [27]. The error bars represent the standard errors of the average 4cB12. Differences between groups were analyzed using one-way ANOVA, while differences between SU and non-SU were computed by Student´s t-test; * *p* ≤ 0.05. OMN = omnivores, LOV = lacto-ovo-vegetarians, VEG = vegans, SU = supplement users, non-SU = non-supplement users, and 4cB12 = four marker combined vitamin B-12 indicator.

**Table 1 nutrients-11-01146-t001:** Participant characteristics by dietary patterns of the study population.

Measure	Omnivores (*n* = 27)	Lacto-Ovo (*n* = 26)	Vegan (*n* = 28)	*p* value
Age, years	27.4 ± 4.03	27.6 ± 4.31	27.5 ± 4.24	0.968 ^a^
Sex	m = 11, f = 16	m = 10, f = 16	m = 10, f = 18	0.929 ^b^
BMI, kg/m^2^	22.3 ± 1.74	21.6 ± 1.98	22.1 ± 2.09	0.436 ^a^
Duration of diet				0.001 ^b^
0.5–1 year, *n* (%)	0 (0)	4 (15)	6 (21)	
1–2 years, *n* (%)	1 (4)	3 (12)	4 (14)	
2–3 years, *n* (%)	0 (0)	2 (8)	7 (25)	
>3 years, *n* (%)	26 (96)	17 (65)	11 (39)	
Vitamin B_12_ SU, *n* (%)	4 (19)	4 (15)	15 (54)	0.005 ^b^
Vitamin D SU, *n* (%)	5 (22)	1 (4)	7 (25)	0.078 ^b^
Folate SU, *n* (%)	3 (11)	1 (4)	5 (18)	0.262 ^b^
Iron SU, *n* (%)	3 (11)	4 (15)	5 (18)	0.689 ^b^
Calcium SU, *n* (%)	3 (11)	1 (4)	2 (7)	0.210 ^b^
Zinc SU, *n* (%)	4 (15)	3 (12)	2 (7)	0.662 ^b^
Magnesium SU, *n* (%)	5 (22)	4 (15)	5 (18)	0.770 ^b^
Training frequency per week	3.04 ± 0.98	3.24 ± 0.88	3.00 ± 0.85	0.502 ^a^
Running time per week, h	2.72 ± 1.11	3.38 ± 1.43	2.65 ± 1.38	0.079 ^b^

SU = supplement users. Values are given as means ± SD or *n* (%). ^a^ Kruskal Wallis test and ^b^ chi-square test.

**Table 2 nutrients-11-01146-t002:** Biomarkers of Vitamin B_12_ status.

Parameter	Supplementation	Omnivores *n* = 27	*p* value Omnivores vs. Lacto-Ovo	Lacto-Ovo *n* = 26	*p* value Lacto-Ovo vs. Vegan	Vegan *n* = 28	*p* value Omnivores vs. Vegan	*p* value
	n_SU_	5		4		15		
	n_non-SU_	22		22		13		
Vitamin B_12_, pmol/L		323 ± 121	-	316 ± 146	-	320 ± 247	-	0.586 ^b^
	SU	350 ± 112	-	261 ± 149	-	396 ± 318	-	0.590 ^b^
	non-SU	316 ± 124	-	326 ± 148	-	244 ± 115	-	0.118 ^b^
Deficient (<150 pmol/L), *n* (%)		1 (4)		2 (8)		3 (11)		0.349 ^d^
	SU	0 (0)		1 (4)		1 (4)		
	non-SU	1 (4)		1 (4)		2 (7)		
Holo-TC, pmol/L		80.4 ± 30.1	-	85.9 ± 36.9		67.8 ± 39.4	-	0.168 ^a^
	SU	92.4 ± 37.7	-	80.5 ± 53.5		82.0 ± 37.9	-	0.871 ^a^
	non-SU	76.1 ± 28.9	n.s.	86.8 ± 34.7	0.013 ^c^	52.1 ± 37.9	n.s.	0.016 ^a^
Deficient (<35 pmol/L), *n* (%)		1 (4)		2 (8)		6 (21)		0.043 ^d^
	SU	0 (0)		1 (4)		1 (4)		
	non-SU	1 (4)		1 (4)		5 (18)		
MMA, nmol/L		264 ± 174	-	266 ± 176	-	363 ± 570	-	0.693 ^b^
	SU	261 ± 177	-	400 ± 362	-	216 ± 161	-	0.186 ^b^
	non-SU	264 ± 177	-	234 ± 123	-	535 ± 801	-	0.226 ^b^
Deficient (>271 nmol/L), *n* (%)		6 (22)		7 (27)		8 (29)		0.720 ^d^
	SU	1 (4)		2 (8)		1 (4)		
	non-SU	5 (19)		5 (19)		7 (25)		
tHcy, µmol/L		12.2 ± 2.93	-	13.2 ± 6.47	-	12.8 ± 4.26	-	0.920 ^b^
>10 µmol/L, *n* (%)		19 (82)		15 (58)		22 (79)	-	0.266 ^d^
4cB12		0.91 ± 0.50	-	0.91 ± 0.75	-	0.70 ± 0.76	-	0.442 ^a^
	SU	1.12 ± 0.56	-	0.52 ± 1.37	-	1.10 ± 0.67	-	0.490 ^a^
	non-SU	0.86 ± 0.49	n.s.	0.98 ± 0.60	0.020 ^c^	0.35 ± 0.75	n.s.	0.021 ^a^

SU = supplement users, non-SU = non-supplement users, Holo-TC = holotranscobalamin, MMA = methylmalonic acid, 4cB12 = four marker combined vitamin B-12 indicator [27], n.s. = not significant, and tHcy = total homocysteine. Values are given as means ± SD or *n* (%) of the study population in the different cutoff values. ^a^ One-way ANOVA, ^b^ Kruskal Wallis test, ^c^ post hoc test, and ^d^ chi-square test.

**Table 3 nutrients-11-01146-t003:** Biomarkers of folate and vitamin D status.

Parameter		Omnivores *n* = 27	Lacto-Ovo *n* = 26	Vegan *n* = 28	*p* value
	n_SU_	3	1	5	
	n_non-SU_	24	25	23	
RBC folate, nmol/L		2213 ± 444	2236 ± 596	2354 ± 639	0.577 ^a^
	SU	2254 ± 776	1456 ± 0	2903 ± 494	0.134 ^a^
	non-SU	2207 ± 413	2246 ± 586	2233 ± 609	0.966 ^a^
Deficient (<340 nmol/L), *n* (%)		0	0	0	-
	n_SU_	6	1	7	
	n_non-SU_	21	25	21	
25(OH)D, nmol/L		90.6 ± 32.1	76.8 ± 33.7	86.2 ± 39.5	0.354 ^a^
	SU	120 ± 40.4	152 ± 0	117 ± 26.3	0.619 ^a^
	non-SU	82.2 ± 24.6	73.8 ± 30.6	73.8 ± 37.3	0.592 ^a^
Optimal		18 (67)	11 (42)	16 (57)	0.219 ^b^
(≥75 nmol), *n* (%)	SU	5 (19)	1 (4)	7 (25)	
	non-SU	13 (48)	10 (39)	9 (32)	
Sufficient		6 (22)	10 (39)	5 (18)	
(50–74.9 nmol/L), *n* (%)	SU	1 (4)	0 (0)	1 (4)	
	non-SU	5 (19)	10 (39)	4 (14)	
Insufficient		3 (11)	5 (19)	5 (18)	
(25–49.9 nmol/L), *n* (%)	SU	0 (0)	0 (0)	0 (0)	
	non-SU	3 (11)	5 (19)	5 (18)	
Deficient		0 (0)	0 (0)	2 (7)	
(<25 nmol/L), *n* (%)	SU	0 (0)	0 (0)	0 (0)	
	non-SU	0 (0)	0 (0)	2 (7)	

n_SU_ = number of supplement users, n_non-SU_ = number of non-supplement users, RBC folate = red blood cell folate, and 25(OH)D = 25-hydroxyvitamin D. Values are given as means ± SD or *n* (%) of the study population at the different cutoff values. ^a^ One-way ANOVA, and ^b^ Chi-square test.

**Table 4 nutrients-11-01146-t004:** Biomarkers of iron status and hematological parameters.

Parameter		Omnivores *n* = 27,	*p* value Omnivores vs. Lacto-Ovo	Lacto-Ovo *n* = 26,	*p* value Lacto-Ovo vs. Vegan	Vegan *n* = 28	*p* value Omnivores vs. Vegan	*p* value
	n_SU_ n_non-SU_	3 24		4 22		5 23		
Iron serum, µmol/L	f m	14.5 ± 7.91 22.2 ± 6.37	- -	16.7 ± 7.03 20.0 ± 8.76	- -	15.7 ± 6.00 18.4 ± 6.80	- -	0.671 ^a^ 0.493 ^a^
Deficiency (<10 µmol/L), *n* (%)	f m	7 (26) 0 (0)		4 (15) 1 (4)		2 (7) 0 (0)		0.353 ^d^ -
Ferritin, µg/L Depleted iron stores (<15 µg/L), *n* (%)	f m f m	25.9 ± 23.0 115 ± 44.8 7 (26) 0 (0)	- 0.024 ^c^	23.9 ± 12.9 64.3 ± 40.7 6 (23) 0 (0)	- n.s.	32.1 ± 22.8 64.6 ± 36.7 5 (18) 0 (0)	- 0.028 ^c^	0.441 ^b^ 0.010 ^b^ 0.619 ^d^ -
Transferrin, µmol/L Increased iron requirement (≥47.7 µmol/L), *n* (%)	f m f m	46.4 ± 12.9 34.7 ± 4.27 6 (22) 0 (0)	- -	41.6 ± 7.79 38.2 ± 4.40 3 (12) 1 (4)	- -	40.2 ± 7.54 39.0 ± 6.16 3 (11) 0 (0)	- -	0.316 ^b^ 0.092 ^b^ 0.306 ^d^ -
Transferrin saturation	f m	17.2 ± 12.6 32.5 ± 10.2	- -	21.0 ± 9.97 26.8 ± 12.2	- -	20.4 ± 8.93 24.8 ± 11.6	- -	0.543 ^a^ 0.288 ^a^
Insufficient iron supply (<16%), *n* (%)	f m	10 (37) 0 (0)		5 (19) 1 (4)		7 (25) 3 (11)		0.184 ^d^ 0.261 ^d^
Hb, g/dL	f m	13.0 ± 1.08 15.1 ± 0.64	- -	13.6 ± 0.78 14.8 ± 1.00	- -	13.4 ± 1.20 15.2 ± 0.84	- -	0.260 ^a^ 0.661 ^a^
Anemia (<12.0 g/dL), *n* (%) (<13.0 g/dL), *n* (%)	f m	3 (11) 0 (0)		0 (0) 0 (0)		4 (14) 0 (0)		0.198 ^d^ -
Hct, L/L	f m	0.39 ± 0.03 0.44 ± 0.03	- -	0.41 ± 0.03 0.43 ± 0.02	- -	0.41 ± 0.04 0.44 ± 0.03	- -	0.083 ^a^ 0.730 ^a^
< 0.36 (f)/0.39 (m), *n* (%)		0 (0)		0 (0)		0 (0)		-
MCV, fl Iron deficiency anemia (<80 fl), *n* (%)	f m	87.7 ± 3.34 87.2 ± 2.78 0 (0)	- -	89.3 ± 5.06 88.8 ± 3.42 0 (0)	- -	89.5 ± 3.59 87.3 ± 4.05 0 (0)	- -	0.410 ^b^ 0.597 ^b^ -

MCV = Mean corpuscular volume. Values are given as means ± SD or *n* (%) of the population at the different cutoff values. ^a^ One-way ANOVA, ^b^ Kruskal Wallis test, ^c^ post hoc test, and ^d^ chi-square test.

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
