# Peer review of "Micronutrient Status of Recreational Runners with Vegetarian or Non-Vegetarian Dietary Patterns"

_nutrients, 2019, doi:10.3390/nu11051146_

Round 1

Reviewer 1 Report

This article explores a really actual topic that still needs data. The design of the study is good and the results are of interest. Plant-based diets are increasingly adopted not only for ethic reasons but also to ameliorate outcomes in selected sports and diseases. The comprehension of nutritional aspects derived from these dietary habits could give practical tools for sustainable planning of food and supplements. However, there may be a large diversion between recreational e professional sports. Nonetheless, this is a topic that greatly needs data to understand the link between sport and nutrition in the exclusion of animal foodstuff.

However, in my opinion, the manuscript needs some improvement. See below some suggestions:   

-        It is not a good idea to use a retracted position as a reference [7]. I think it is better to use updated ones like “J Acad Nutr Diet. 2016;116:1970-1980” and/or “Nutr Metab Cardiovasc Dis. 2017 Dec;27(12):1037-1052” and/or “Medicine & Science in Sports & Exercise. 48(3):543–568, MAR 2016” and/or “ Journal of the International Society of Sports Nutrition 2017 14:20” and/or “J Int Soc Sports Nutr. 2017; 14: 36.”

-        The categorization of diets is not so clear. In methods, it is described that the categorization of the 3 diets was based on questionnaires including FFQ but there is no information about the cut-offs of frequencies used to lie into one category than another. This is a major clue in categorization. This aspect needs to be better clarified.

-        The 4cB12 is a good tool to discriminate among blood level artefacts that hide a subclinical shortage. Even if the reference has been provided, could be useful to briefly describe the method, at least the formula and the sample population for the validation.

-        Speaking about Calcium, it must be discussing that blood levels are not directly linked to the intake and that a low Ca blood concentration appears only in the worse deficiency condition. Usually, calcium has an electrolyte function significance. Moreover, data on parathormone could be useful.

-        Is zinc blood concentration reliable to define an adequate intake in the diet?

-        At lines 135-140, ranges of biomarkers are taken form clinical diagnostic manuals (with pages shown in the text). So, why there are multiple references? I think that it must be shown only the one reference for each range. For example, [51] for Calcium; [53] for Magnesium; [54] for Zinc and [39] for the rest (because it refers to WHO cut-offs).

-        About supplement users, it is important to specify frequency and dosage, in particular with vitamin B12. Cobalamin absorption for single dose ingestion is greatly weakened by the limit of active transport. So, low dosages can be of minor influence in the stabilization of blood markers.

-        About folate, it must be stated that the population sample does not pertain to a mandatory fortification area.

-        At line 218, p=184 refers to transferrin saturation and not to ferritin levels.

-        About male/female in table 4, I think there are some errors in the table alignment that make difficult the interpretation.

-        At lines 256-258, the statement is not in line with the literature that considers a LOV diet inadequate for the vitamin B12, without a supplementation plan. Data presented regarding LOV about vitamin B12 are weak to make this statement, taking also in account that the duration of the diet is crucial for the breakdown of the body stocks. 

-        In the conclusion section, the limitations of the study must be provided to limit the overinterpretation of the results.

Author Response

Point-by-point response to the comments made by the reviewers

We thank the reviewers for their kind comments and the opportunity to revise our paper. The comments and suggestions have helped to improve the manuscript substantially. In the following, each comment is addressed separately.

Reviewers’ comments:

Reviewer 1:

This article explores a really actual topic that still needs data. The design of the study is good and the results are of interest. Plant-based diets are increasingly adopted not only for ethic reasons but also to ameliorate outcomes in selected sports and diseases. The comprehension of nutritional aspects derived from these dietary habits could give practical tools for sustainable planning of food and supplements. However, there may be a large diversion between recreational e professional sports. Nonetheless, this is a topic that greatly needs data to understand the link between sport and nutrition in the exclusion of animal foodstuff.

-        It is not a good idea to use a retracted position as a reference [7]. I think it is better to use updated ones like “J Acad Nutr Diet. 2016;116:1970-1980” and/or “Nutr Metab Cardiovasc Dis. 2017 Dec;27(12):1037-1052” and/or “Medicine & Science in Sports & Exercise. 48(3):543–568, MAR 2016” and/or “ Journal of the International Society of Sports Nutrition 2017 14:20” and/or “J Int Soc Sports Nutr. 2017; 14: 36.”

Response: Thank you for your thorough review. We have deleted the reference and added the following: “J Acad Nutr Diet. 2016;116:1970-1980”, “Nutr Metab Cardiovasc Dis. 2017 Dec;27(12):1037-1052” and “Medicine & Science in Sports & Exercise. 48(3):543–568, MAR 2016”.

-        The categorization of diets is not so clear. In methods, it is described that the categorization of the 3 diets was based on questionnaires including FFQ but there is no information about the cut-offs of frequencies used to lie into one category than another. This is a major clue in categorization. This aspect needs to be better clarified.

Response: Thanks for this hint. We have described the method of categorization in more detail as follows: “The categorization in omnivorous, lacto-ovo-vegetarian and vegan was based on questionnaires which initially included a question about the current diet. Secondly, consumed food groups were queried to make sure that the participants classified themselves correctly. Subjects were classified as “omnivorous”, if they consumed grains, plant foods, legumes, milk and dairy products, eggs, as well as fish, meat and meat products. “Lacto-ovo-vegetarians” have been characterized by the consumption of grains, plant foods, legumes, milk and dairy products, and eggs. The consumption of grains, plant foods, and legumes characterized “vegans”.”

-        The 4cB12 is a good tool to discriminate among blood level artefacts that hide a subclinical shortage. Even if the reference has been provided, could be useful to briefly describe the method, at least the formula and the sample population for the validation.

Response: We complemented the formula and briefly described the discussion as follows: “The formula for calculating 4cB12 was established with a database of 5211 subjects from various nations, ages and health status [27].”

-        Speaking about Calcium, it must be discussing that blood levels are not directly linked to the intake and that a low Ca blood concentration appears only in the worse deficiency condition. Usually, calcium has an electrolyte function significance. Moreover, data on parathormone could be useful.

Response: We added a short part in the discussion about calcium, zinc and magnesium (4.5.) as follows: “Average values of all three groups showed adequate serum levels of all three minerals. Most subjects of VEG had low levels of zinc, which is consistent with previous findings [14]. However, in contrast to the previously mentioned parameters, serum levels of calcium, zinc and magnesium are not directly linked to the dietary intake, due to the tight homeostatic regulation of blood levels.” Moreover, we changed the term “Biomarkers” to “Serum levels” of calcium, zinc, and magnesium. Unfortunately, we don’t have data on PTH levels.

-        Is zinc blood concentration reliable to define an adequate intake in the diet?

Response: Similar to calcium and magnesium, zinc is also subject to strict homeostatic regulation and therefore not directly reflects its absorption. We added this aspect in the discussion (see the previous comment).

-        At lines 135-140, ranges of biomarkers are taken form clinical diagnostic manuals (with pages shown in the text). So, why there are multiple references? I think that it must be shown only the one reference for each range. For example, [51] for Calcium; [53] for Magnesium; [54] for Zinc and [39] for the rest (because it refers to WHO cut-offs).

Response: We adjusted the references and specified only one reference for the mentioned parameters.

-        About supplement users, it is important to specify frequency and dosage, in particular with vitamin B12. Cobalamin absorption for single dose ingestion is greatly weakened by the limit of active transport. So, low dosages can be of minor influence in the stabilization of blood markers.

Response: We agree with the reviewer. We calculated the correlation of the daily vitamin B12-supplement dosage to 4cB12 as vitamin B12 status marker. Interestingly, in the VEG group, correlations between supplementation and 4cB12 were found (p=0.025), while associations were neither found in the OMN and LOV group nor in the whole study population.

Since the vitamin D status was also dependent on supplementation, we calculated correlations for the individual groups. We found an association for the OMN (p=0.039) and VEG group (p=0.001). In contrast, no significant correlations were found in LOV (p=0.127).

For the sake of completeness, we also conducted correlation analysis between the iron supplement intake and biomarkers of iron status, but found no associations.

We integrated the correlation of the daily supplement intake and the biomarkers for vitamin B12 and vitamin D in the result section.

-        About folate, it must be stated that the population sample does not pertain to a mandatory fortification area.

Response: We have added that in the discussion.

-        At line 218, p=184 refers to transferrin saturation and not to ferritin levels.

Response: Thank you for this hint. We corrected the p value.

-        About male/female in table 4, I think there are some errors in the table alignment that make difficult the interpretation.

Response: We are sorry if there have been shifts here. However, we cannot see these shifts. In our table everything is depicted correctly. Maybe there is a formatting error?

-        At lines 256-258, the statement is not in line with the literature that considers a LOV diet inadequate for the vitamin B12, without a supplementation plan. Data presented regarding LOV about vitamin B12 are weak to make this statement, taking also in account that the duration of the diet is crucial for the breakdown of the body stocks.

Response: We formulated the sentence with more caution and integrated the aspect of the duration of the diet as follows: “Considering the non-SU, it is obvious that VEG had the lowest vitamin B12 supply markers and as the duration of the vegan diet increased, there was a trend towards a lower status. Interestingly, there were only marginal differences between the non-SU of the OMN and LOV group suggesting that a lacto-ovo-vegetarian diet, containing milk and dairy products, provides certain amounts of vitamin B12. In addition, the duration of the diet had no impact on the vitamin B12 supply in OMN and LOV. Nevertheless, our results should be considered cautiously, as these findings are in contrast to most studies showing an inadequate B12 status in the majority of vegans and vegetarians as summarized in [56].”

Additionally, we included the results of the new calculations in the result section.

-        In the conclusion section, the limitations of the study must be provided to limit the overinterpretation of the results.

Response: We included a limitation part (4.6.).

Reviewer 2 Report

The manuscript of Nebl J et al describes the micronutrient status of runners related to the type of it diet (OMN, LVO, VEG). Based on this cross-sectional study was concluded that well planed, lacto-ovo-vegetearian diet, including supplements, can meet the athlete's needs of vit B12, vitamin D and iron.

This research has some limitations that should be clearly highlighted in the manuscript. First of all  small amount of runners involved in the study. Second, only 18-35 y.o. runners were included. Third, the obtained results may vary depends on season, type of physical activity, climate etc. That's why I recommend to add a paragraph about study limitations.

In overall I like this manuscript.

Author Response

Point-by-point response to the comments made by the reviewers

We thank the reviewers for their kind comments and the opportunity to revise our paper. The comments and suggestions have helped to improve the manuscript substantially. In the following, each comment is addressed separately.

Reviewers’ comments:

Reviewer 2

The manuscript of Nebl J et al describes the micronutrient status of runners related to the type of it diet (OMN, LVO, VEG). Based on this cross-sectional study was concluded that well planed, lacto-ovo-vegetearian diet, including supplements, can meet the athlete's needs of vit B12, vitamin D and iron.

This research has some limitations that should be clearly highlighted in the manuscript. First of all  small amount of runners involved in the study. Second, only 18-35 y.o. runners were included. Third, the obtained results may vary depends on season, type of physical activity, climate etc. That's why I recommend to add a paragraph about study limitations.

In overall I like this manuscript.

Response: That’s an important point. We included a limitation part (4.6.) at which we entered the small sample size and the influence of the season and current training phase.

Round 2

Reviewer 1 Report

The Authors accurately reviewed the manuscript according to the suggestion and improved the results with other informations.